# Determining Reproductive Parameters, which Contribute to Variation in Yield of Olive Trees from Different Cultivars, Irrigation Regimes, Age and Location

**DOI:** 10.3390/plants11182414

**Published:** 2022-09-16

**Authors:** Tahel Wechsler, Ortal Bakhshian, Chaim Engelen, Arnon Dag, Giora Ben-Ari, Alon Samach

**Affiliations:** 1The Robert H. Smith Institute of Plant Sciences and Genetics in Agriculture, The Robert H. Smith Faculty of Agriculture, Food and Environment, The Hebrew University of Jerusalem, P.O. Box 12, Rehovot 76100, Israel; 2Gilat Research Center, Agricultural Research Organization (ARO), Volcani Center, Gilat, Mobile Post Negev 2, Negev 85280, Israel; 3Institute of Plant Sciences, Agricultural Research Organization (ARO), Volcani Center, Rishon LeZion 7528809, Israel

**Keywords:** age, juvenility, flowering, fruit set, fruit production, rainfed, *Olea europaea*, olive oil

## Abstract

Olive (*Olea europaea* L.) trees can reach a very old age and still bear fruit. Although traditional groves are planted at low density and are rainfed, many newer groves are planted at higher densities and irrigated. As expected, initial yields per area are larger in high density plantations, yet some farmers claim they experience a reduction in productivity with grove age, even in well maintained trees. In order to test the accuracy of this claim and its underlying cause, we measured several productivity parameters in selected branches of trees in seven sites differing in cultivar (‘Barnea’ or ‘Souri’), location and irrigation regime (rainfed or irrigated) for two consecutive years. For each site (cultivar/location/regime), we compared neighboring groves of different ages, altogether 14 groves. There was no consistent reduction in productivity in older groves. Differences in productivity between irrigated cultivars were mostly due to variation in the percentage of inflorescences that formed fruit. Several parameters were higher in irrigated, compared to rainfed ‘Souri’. Differences in productivity between years within the same grove was mostly due to variation in the percentage of nodes forming inflorescences. We studied the expression of *OeFT2* encoding a FLOWERING LOCUS T protein involved in olive flower induction in leaves of trees of different ages, including juvenile seedlings. Expression increased during winter in mature trees and correlated with the percentage of inflorescences formed. The leaves of juvenile seedlings expressed higher levels of two genes encoding APETALA2-like proteins, potential inhibitors of *OeFT2* expression. The buds of juvenile seedlings expressed higher levels of *OeTFL1*, encoding a TERMINAL FLOWER 1 protein, a potential inhibitor of OeFT2 function in the meristem. Our results suggest that olives, once past the juvenile phase, can retain a similar level of productivity even in densely planted well maintained groves.

## 1. Introduction

Olive (*Olea europaea* L.) is an evergreen tree from the Oleaceae family, originating in the Mediterranean basin [1]). Olive oil and table olives are a huge commodity in the world markets. Olive oil is associated with the healthy Mediterranean Diet, increasing its demand [2,3]. The world produces about 3.2 million tons of olive oil every year, approximately 66% in Mediterranean EU countries. EU countries consume approximately 50% of produced oil (INTERNATIONAL OLIVE OIL COUNCIL website https://www.internationaloliveoil.org/, accessed on August 2022).

Olive cultivation in the Middle East may have begun as far back as 6500 years ago [4]. Olive trees are considered long living, with evidence suggesting some trees reach an age of over 600 years [5]. Furthermore, in the Garden of Gethsemane in Jerusalem, the age of three olive trees was estimated at 800–900 years [6], with these trees continuing to bear fruit. Other old trees in Israel are likely to be at least 200–300 years old [7]. Thus, well-treated rainfed olives can continue to bear fruit for hundreds of years. Most commercial olive orchards grow specific olive varieties that are vegetatively propagated by rooted cuttings. Trees from each variety are genetically identical and no longer require a long transition from the juvenile to mature phase (see below). These olive trees flower once every year in the spring. Still, full production from these trees may take several years.

When propagated from seeds, olive seedlings cannot flower in the first years (up to 10–15 years) even under favorable environmental conditions. The stage is termed as the ‘juvenile phase’. Once a plant reaches an age in which it can flower in response to environmental or internal cues, it is considered to have transitioned to the ‘mature phase’. In annuals, the juvenile phase may last a few days, while in some woody perennials it can last several years [8]. Olives grown from seedlings have a relatively long juvenile phase. For example, in one report, under field conditions, only 21% of seedlings reached the mature phase by the age of 14 years [9].

It appears that the proteins and microRNAs involved in the transition from juvenility to maturity are conserved in different plant species. In Arabidopsis, the expression of a family of APETALA2 (AP2)-like transcription factors, with an ability to repress flowering, is reduced with the transition [10]. These genes are targeted by miR172 microRNAs that increase with age during the transition [10,11]. AP2-like proteins repress flowering by inhibiting the expression of some genes involved in flowering, such as FLOWERING LOCUS T (FT) and SUPPRESOR OF CONSTANS1 (SOC1) [12]. The *Olea europaea* genome [13,14] provides an opportunity to identify similar proteins in olives and to use uniform nomenclature for olive genes isolated in previous research. The olive Oeu-mir172, as well as two AP2-like genes with target sites (contig #15488; currently OE6A99997 and contig#6113.6; OE6A079258) were identified [15]. In olive, during the transition to maturity, there is evidence supporting a decrease in the expression in shoot apices of a gene encoding a protein with an AP2 domain (contig # 5004; OE6A119978; [16]).

In regions with relatively cold winters, such as Israel, during the winter there is normally a cessation of olive vegetative growth. As temperatures warm up towards the end of winter, vegetative growth begins and olive inflorescences elongate from some of the axillary buds. In most varieties, the apical meristem remains vegetative, and much of the new growth continues from this meristem. There is also new growth emerging from some vegetative meristems formed in olive axillary buds. In the spring, vegetative growth is vigorous and in conditions of hot summers and existence of developing fruits acting as a sink, there can be a decrease in the vegetative growth rate [17]. In the fall, depending on olive fruit load and availability of water and nutrients in the soil, there may be a moderate increase in vegetative growth rate. In areas with warmer winters, growth can even continue during winter [18].

Olive inflorescences form within lateral buds of one-year-old shoots. A major factor that affects olive production is the number of inflorescences emerging during the spring. This value depends on two parameters: the number of new axillary buds formed between the previous spring and autumn (*n*) and the percentage of these buds that actually go through the flowering transition and emerge in spring (*i*; ranging from 0% to 95%). Both *n* and *i* are affected by the previous year’s fruit load. Under heavy fruit load vegetative growth will be inhibited, leading to shorter new branches with fewer nodes and axillary buds [17]. In addition, in the buds that do form, the ‘memory’ of fruit load inhibits flower induction [19]. Due to this combined effect, a year with a heavy crop (ON year) will lead to a year with little flowering and fruiting the following year (OFF year), a phenomenon termed ‘Alternate Bearing’. Alternate bearing is a common phenomenon in almost all species of fruit trees: deciduous, evergreen and subtropical [20]. The amplitude of alternate bearing in olives normally increases with tree age [21] when trees in an orchard become synchronized by extreme environmental conditions that may harm flowers or developing fruitlets, leading to an OFF year. Flower induction in olives requires low winter temperatures [19]. An increase in the expression of genes encoding the flower-promoting FT protein occurs during winter (December–January in Israel) in response to low temperatures, and the level of *FT* mRNA increases, depending on the level of winter chilling, yet it is negatively affected by the memory of previous fruit load [19]. As a result, in order to reach the same level of flowering, trees that experienced a higher fruit load will need to be exposed to a cooler winter. This is true to a certain point, since after a very heavy fruit load, flowering will be low, even if the winter was optimal for flower induction.

Under local conditions in Israel, in the olive cv. ‘Barnea’, microscopic inflorescences are formed in lateral meristems towards the end of February [19], they later emerge and elongate and flowers reach anthesis towards the end of April. Each inflorescence has 10–35 male or hermaphroditic flowers. Olives are self-incompatible [22] and wind pollinated. Not all hermaphroditic flowers will go through fertilization, and some of the developing fruitlets will abscise at an early stage of fruit development, mostly within 20 days after anthesis [23]. On average, only 0.1–0.7 fruits per inflorescence will survive until the fruits ripen [24]. In Israel, table olives are harvested in August, and olives for oil are harvested in late autumn and early winter (October–November).

Traditional olive orchards in the Mediterranean basin are rainfed and planted in low density (70–125 trees/ha) [25]. Without ‘rejuvenation’ techniques, such as pruning, the capacity of older trees to produce sufficient one-year-old shoots declines [26]. Several studies on the positive effect of pruning on older olive trees have been published [27,28,29]. In Israel, approximately 24% of olive groves are irrigated, producing 66% of the total yield [30]. Among the irrigated orchards, most are grown at an average density termed ‘intensive’ (200–400 trees/ha) with fruits harvested using different types of shakers [25,31]. The move to irrigated intensive groves began relatively early in Israel [32,33], thus there are many such groves that are more than 20 years old. Recently, many local farmers developed the impression that the profitability of regularly pruned, irrigated olive groves is limited in time. According to them, over time, and perhaps due to the intensive growth, there has been a decrease in yields, and, therefore, the lifespan of a commercial irrigated orchard is limited. To test this, we studied two different age groups of well-managed groves at seven sites. In each grove, for two consecutive years, we measured *n*, *i*, the number of inflorescences per branch (*k*), the percentage of inflorescences that kept fruit (*f*), the average number of fruits on those inflorescences (*m*) and the average number of final fruits on the branch (*x*). Three sites were of irrigated ‘Barnea’, two sites of irrigated ‘Souri’ and two sites of rainfed ‘Souri’. Based on these measurements, we identified the main parameters affecting the yield of these orchards and tested the effect of age, cultivar and irrigation regime on these parameters. In addition, the expression of select flowering time genes was monitored in tissues of different aged trees.

## 2. Materials and Methods

### 2.1. Plant Material

The experimental design to study the effect of age on productivity was a paired comparison design. We selected sites in which two close-by groves of different age, from the same variety, were grown under identical irrigation conditions and similar farm practices. In the spring of 2016, we began the experiment in nine sites. In the summer of 2017, two of the nine sites were unavailable for continued experimentation. One was uprooted, and one of the rainfed plots started to receive irrigation. This led to the removal of these sites from our study, leaving us with seven sites, which serve as replicates, each site containing two groves of different age (Table 1). Three sites were ‘Barnea’-irrigated groves. Two sites were ‘Souri’-irrigated groves, and two other sites were ‘Souri’-rain-fed groves (Table 1). Data on variety and planting year were obtained from the farmers, and in Rame, the date of the older grove was a very rough estimate. Based on these estimates, the age differences between trees within each plot were between 7 to 950 years. The density of the trees used in intensive irrigation (harvesting with the help of a shaker) is approximately 400 trees/ha (spacing of 7 m × 4 m between trees in a row), and in the rainfed plots it was 200 trees/ha (Yodfat, 7 m × 7 m) or 100 trees/ha (Rame, 10 m × 10 m). Pruning is conducted every winter. The amount of irrigation in irrigated groves is normally calculated based on approximately 80% of potential crop evapotranspiration (in accordance to measurements from nearby meteorological station) multiplied by a cover factor estimated at 50% in adult orchards, reaching 4000 m^3^ ha^−1^ y^−1^. Fertigation in these orchards depends on water quality, at rates per year of 35–150 kg N ha^−1^, 0–50 kg ha^−1^·P and 0–150 kg ha^−1^ K. In rainfed organic groves, in the beginning of winter, once every two years, 60–80 m^3^ ha^−1^ compost is added. In other rainfed groves, 3–5 kg ammonium sulphate ((NH_4_)_2_SO_4_) 0.7 kg phosphorus pentoxide (P_2_O_5_) and 1.5–2 kg potassium chlorate (KCl) are added to every tree, every year. Those are the recommendations of the extension service of the ministry of agriculture in Israel.

### 2.2. Trees and Branches Used for Measurements

We conducted the experiment until the winter of 2018, using the same trees and marking new branches every year. In these groves, during the spring–summer of 2016, uniform-looking olive trees with medium levels of flowering were chosen, six trees for each age group, altogether eighty-four trees. On each tree, 25 branches without flowers (if there were few flowers they were removed) were marked with a colored ribbon at the point of new spring growth. From this point onwards, towards the apex, there is formation of new lateral buds with a potential to initiate inflorescences that can reach anthesis in the following spring. Each branch was numbered, so that we could follow its fate in several time points. We avoided marking abnormal branches, such as epicormic shoots.

### 2.3. RNA Sampling

RNA samples were collected from ‘Barnea’ trees of both ages in Tzabar Kama and in Revivim. For each marked tree, on November 16th 2016 (Tzabar Kama) and in mid-January 2017 (both regions), leaves (Figure 1A,B) and stems with axillary buds (Figure 1C) were sampled. At the same dates that the mature trees in the ‘Tzabar Kama’ grove were sampled, we also sampled leaves of juvenile trees (seedlings) (Figure 1E) at the ARO center, Beit Dagan, Israel. Whereas the female parent of these juvenile trees was ‘Barnea’, the male parent is unknown. Sampling was done in the morning, and the tissues were collected from the growth of the last year, which has the potential to produce inflorescences. The process was similar to that published [19]. Two branches were harvested from each tree, and from each branch, in a uniform position, three leaves with a healthy appearance and separately three stems with lateral buds were collected. The plant material, placed in appropriate tubes (Figure 1D), was immediately frozen in liquid nitrogen and transferred to a −80 °C freezer for storage until RNA extraction.

### 2.4. RNA Production and cDNA Synthesis

The real-time PCR protocol through which the expression levels of genes were quantified is according to a recent study [19]. Extraction of total RNA was performed using the improved guanidine method [34]. RNA concentration was determined in a spectrophotometer (ND-1000, NanoDrop Technologies, Rockland, DE, USA). The mRNA was isolated from 7.5 µg of total RNA using magnetic oligo-dT beads (Dynabeads Oligo (dT) _25_, Invitrogen) according to manufacturer’s instructions. First-strand cDNA was synthesized using Superscript II Reverse Transcriptase (Invitrogen) and Oligo (dT) _12-18_ primers.

### 2.5. Expression Analysis by Quantitative Real-Time PCR

Analysis of cDNA samples was performed using the ABsolute Blue QPCR ROX Mix (Thermo Scientific). Reactions were run on a Rotor-Gene 6000 cycler (Corbett Life Science, Sydney, Australia), in two/three technical repeats. Quantification of each gene was performed using Corbett Research Rotor-Gene Software, as previously described [35]. Primers for real-time reactions for the genes are presented in Table 2. *OeFT2* (OE6A103537), *OeTFL1-1* (Oe6A037908) and *OeACT7* (OE6A117728) primers were previously used [19].

### 2.6. Reproductive Parameters Collected on Selected Branches

In both April 2017 and April 2018, *n* and *k* values were measured and *i* was calculated for each of the 2100 marked branches in the experiment. Branch length in cm was not measured, assuming that *n* values are more relevant for reproductive potential. Average *n*, *k* and *i* values were calculated for each tree, based on data from 25 branches. In June 2017 and June 2018, after natural fruitlet abscission, a survey of remaining fruitlets was conducted in each of the marked branches. We counted how many inflorescences kept fruit and calculated *f*. We counted how many fruits remained on each branch (*x*) and calculated *m*. Average *f*, *x* and *m* values were calculated for each tree, based on data from 25 branches.

In 2018, we carried out a full harvest of 17 experimental trees in three groves in Revivim. The trees were from two age groups in ‘Souri’ (harvested in October) and from the older age group of ‘Barnea’ (harvested in November). Harvesting was done manually with the help of long handled combs; the olives were harvested on a net and collected into cloth bags. Once harvested, the fruit yield in kg per tree was measured in the field using a portable hanging weight (±10 g). We also sampled 30 fruits per tree, weighed them using a sensitive scale in the laboratory and calculated average fruit weight per tree. By dividing total fruit weight with average fruit weight for each tree, we estimated the number of fruits harvested per tree (*Y*).

### 2.7. Calculation of Alternate Bearing (AB) Index

The degree of alternate bearing (AB) can be measured using different parameters: yield, total hermaphrodite flowers, total flowers or *i*. *i* is strongly affected by previous fruit load [19]. The AB index measures differences in *i* of the same trees between two consecutive years. The index ranges from 0 (similar to *i* in both years) to 1 (flowering in only one of the years), using the equation:$$\mathrm{AB}\;\left(i\right)index=\frac{|i^{year\;1}-i^{year\;2}|}{i^{year\;1}+i^{year\;2}\;}$$

### 2.8. Statistical Analysis

Data were analyzed by one-way analysis of variance (ANOVA) using JMP Pro 14 software (SAS Institute, Cary, NC, USA). Differences between treatments were determined by Student’s *t* test. In multiple comparisons, Tukey–Kramer honestly significant difference (HSD) was implemented. Statistical significance was determined at *p* ≤ 0.05. In cases where the variance was unequal and the data did not show a normal distribution, the statistical tests were conducted on the transformation to ranks of the values.

## 3. Results

### 3.1. An Estimate of Total Fruit Yield Based on 25 Pre-Marked Branches

The experiment included fourteen groves. For each grove, data were collected from six trees, which had been characterized as having an intermediate level of flowering the previous year. Every year, data were collected for each tree from 25 pre-marked branches. Initially we asked how well data collected from 25 branches can accurately represent the entire tree. To answer this question, we tested a possible correlation between *x* and total fruit number (*Y*) per tree in 17 trees of different groves in one location (Revivim, Figure 2). The R-squared value was 0.86, suggesting that *x* values averaging 25 randomly pre-chosen branches are highly corelated with the total tree fruit number.

### 3.2. Older Trees do Not Necessarily have Lower Yields

We then asked whether there is any evidence for a reduction in yield caused by age. To avoid the influence of alternate bearing, we also calculated the two-year average of *x* for each plot (Figure 3C). Average values varied from 1.2 to 10 fruits per branch (Figure 3). At five of the seven groves there was a significant difference in *x* between age groups within the grove. Of these five, in three groves the younger plot had more fruit per branch, while in the other two groves the older plot had more fruit. When comparing only irrigated groves, half of the groves in the younger plot had the higher *x* value, while in the other half, the older plot had the higher *x* value. Thus, we found no evidence suggesting a general age-dependent decrease in yield in irrigated well-managed olive groves.

There were very high (0–17.8) fluctuations in *x* between years, plots, varieties and irrigation methods (Figure 3). *x* was significantly higher in irrigated ‘Souri’, compared to rainfed ‘Souri’ trees (Table 3). *x* was also significantly higher in irrigated ‘Barnea’ versus irrigated ‘Souri’ trees (Table 3).

In order to try and explain this variation in *x*, we aimed to identify a similar fluctuation of a different parameter at an earlier phenological stage. We counted *n* and *k* and calculated *i*, *f*, and *m*. *x* can be calculated based on the number of inflorescences and the degree to which those inflorescences developed fruit:$$x=\frac{n\times i}{100}\times \frac{f\times m}{100}$$

Given that the correlation between *x* and *Y* is relatively high (see above), collecting data on *n*, *i*, *f* and *m* should provide an important understanding of the major parameters that affect olive fruit numbers per tree.

### 3.3. Rainfed Trees Form Fewer Buds per Branch (n) Compared to Irrigated Trees

On the scale of the individual tree, per year of the study, *n* varied between 8.8 and 37.7 (data not shown). When studying averages of trees from the same treatment and year, *n* varied between 10.9 and 28.4 (Figure 4A,B). A fruit-bearing branch produces far fewer new nodes than one without any fruit load [17]. It should, therefore, be noted that although we studied the same trees during both study years, in each year we chose to measure branches with no fruit. Still, differences in node number in the same trees, between the two years may be associated to overall fruit load in these trees. For ‘Barnea’ in Ein Hanatziv there was an increase in *n* in the second year in both age groups. In Zabar Kama’s younger ‘Barnea’ plot, there was a decrease in *n* in the second year (Figure 4A,B). The two-year average of *n* varied from 11.4 to 28.2 among the different plots (Figure 4C). When studying the effect of tree age on *n* in the seven sites, in three sites significantly higher n was measured in the younger grove, and, in one site, significantly higher *n* was measured in the older grove (Figure 4C). Irrigation significantly increased (1.6 fold) *n* in ‘Souri’ trees (Table 3). Irrigated ‘Souri’ trees had a significantly higher *n* (1.3 fold), compared to ‘Barnea’ trees (Table 3). This could be a genetic trait of ‘Souri’, yet it also might be an indirect effect since ‘Souri’ trees have lower yields (*x*), thus allowing more vegetative growth.

### 3.4. Variation in I due to Alternate Bearing and OeFT2 Expression in Leaves

*i* is used for measuring the degree of flowering induction in olives [19]. We counted the number of Inflorescences formed on each branch (Appendix A), and calculated *i* values for each branch, based on *k* and *n*. On the scale of the individual tree, per year of the study, *i* varied between 0 and 94% (data not shown). When looking at the averages of trees from the same grove (identical treatment and planting year), *i* still varied from 1 to 91% (Figure 5A,B). *i* varied between 5.4 and 78.3% in irrigated ‘Barnea’, between 25.1 and 67.5% in irrigated ‘Souri’ and between 1.2 and 91% in rainfed ‘Souri’. The two-year *i* averages of trees per treatment varied less, between 20 to 51% (Figure 5C), suggesting that for a certain grove, a year with less flowering was followed by a year with much more flowering, termed ‘alternate bearing’. In three ‘Barnea’ groves (both age groups in Ein Hanatziv and younger grove in Tzabar Kama), a simultaneous year-dependent increase/decrease in *n* (Figure 4) and *i* (Figure 5) caused a pronounced change in *k* values (Appendix A). The degree of alternate bearing (AB) with regard to *i* (Figure 5D) can be evaluated by calculating the AB index (see Section 2 for equation). The AB index can vary in values from 0 (the same degree of flowering in both years) to 1 (flowering in just one of the two years). Although we initially chose trees with medium levels of flowering in the spring of 2016, an index of above 0.6, signifying more than a 4-fold change in flowering between years, was identified in five groves representing both cultivars, both irrigation treatments, and both the younger and older plots (Figure 5D). There was no overall effect of age on *i* (Figure 5C), suggesting that older trees do not have reduced flower induction. In the four irrigated sites in which there was a significant difference between age groups in *x* (Figure 3C), there was also a significant difference between age groups in *i* (Figure 5).

We asked if differences in *i* between age groups are correlated by differences in the expression of the *OeFT2* gene in leaves. We compared relative *OeFT2* gene expression in leaves sampled on January 17th 2017 in four groves of irrigated ‘Barnea’, in Tzabar Kama and Revivim (Figure 6A). *OeFT2* levels were significantly higher in the young, compared to the old grove of Tzabar Kama correlating with *i* (Figure 6B). In Revivim, *OeFT2* levels in the two age groups were not significantly different (Figure 6A), correlating with *i* (Figure 6B). When comparing levels of *OeFT2* expression between different sites at a specific date, *OeFT2* levels were not significantly different between the younger groves, yet *i* levels were much higher in Tzabar Kama. The accumulation of *OeFT2* in leaves depends on cold winter temperatures [19], and the dynamics of accumulation may be affected by temperatures during and before sample collection, which may have been different in the two locations. Thus, comparing *OeFT2* levels from leaves at two different locations at a single timepoint may not predict *i*.

Irrigation did not significantly affect *i* in ‘Souri’, so the significant increase in *k* was a result of irrigation increasing *n*, as well as *f* (Table 3). In irrigated orchards, the ‘Souri’ cultivar showed significantly higher *n* and *i*, compared to ‘Barnea’, leading to significantly lower numbers of *k* in ‘Barnea’ (Table 3). As mentioned above, *n* values were significantly higher in ‘Barnea’ (Table 3), suggesting that reproductive stages after flowering are less efficient in ‘Souri’.

### 3.5. f and m Values Were Significantly Lower in ‘Souri’ Compared to ‘Barnea’

As olive inflorescences contain 10–35 male or hermaphroditic flowers, with wind pollination, even self-incompatible cultivars would potentially have at least one fruit developing from each inflorescence. When looking at the scale of the individual tree in a single year, *f* varied from 0 to 94%. On the treatment scale, the averages of trees from the same treatment and year, the value of *f* varied from 0 to 75% (Figure 7A,B). The two-year averages of trees per treatment varied from 14 to 71% (Figure 7C). No age-dependent effect was detected for *f*. In ‘Souri’, *f* was almost 2-fold higher in irrigated as opposed to rainfed trees (Table 3). Still, *f* levels in ‘Barnea’ were significantly higher than in ‘Souri’ (Table 3). The ‘Barnea’ groves in Tzabar Kama had consistently higher levels of *f* in both years of the experiment. Thus, it appears that *f* is affected by cultivar, is increased with irrigation and can differ among sites. The effect of the site may be through the proximity of nearby pollinizers, as well as other growth conditions that may increase flower quality and fruitlet survival, such as Boron [36] and other minerals [37]. For *m* there was little variability, with inflorescences containing one or two final fruits (Appendix A). Still, cultivar had an impact with ‘Barnea’ having a significantly higher fruit number, compared to ‘Souri’ (Table 3). So, although ‘Souri’ had higher numbers of nodes and higher levels of flowering, the total number of fruits per branch was lower in this cultivar, since less inflorescences held on to fruit and the ones that contained fruit contained less fruits in ‘Souri’ (Table 3). Within ‘Souri’, inflorescences of irrigated trees contained a slightly higher value of *m* (Table 3).

### 3.6. i and f Are the Main Contributors to Variation in x

As mentioned above, the mean number of fruits per branch x, is dependent on the four factors described above, as described in the equation:$$x=\frac{n\times i}{100}\times \frac{f\times m}{100}$$

As described above, the increase in *x* in ‘Barnea’, compared to ‘Souri’ is not a result of increases in each one of the four factors. Studying all the groves, 168 data points were taken from both years of the experiment from 84 trees (each tree provided two data points, one per year), and we looked at what the magnitude of association between each of factors to *x* is, by calculating the coefficient of determination. The values, 0.17, 0.38, 0.29, 0.04 for the *n*, *i*, *f* and *m* indices, respectively, showed little association between a single factor and *x.* The product of two factors, *i* × *f*, showed relatively high association with *x*, with a coefficient of determination reaching 0.84 (Figure 8). This suggests that these two factors combined, explaining most of the variation in fruit numbers per branch.

### 3.7. Juvenile Olive Seedlings Do Not Flower, Likely due to Low Levels of OeFT2 and Higher Levels of Potential Flowering Inhibitors

In this study, we tested whether cultivars planted as vegetative cuttings lose some aspects of reproductivity with time. Our results suggest that there is no overall effect of age of grove on the different variables examined. As mentioned above, olives grown from seedlings have a relatively long juvenile phase, meaning they do not flower even when exposed to inductive cold winters. Indeed, juvenile seedlings grown from seeds of ‘Barnea’ (unknown male parent), also explored in this study, did not flower in the spring of 2017. We compared the expression of flowering-related genes in tissues from juvenile seedlings and ‘Barnea’ trees from Tzabar Kama at two timepoints, at the beginning (17 November) and towards the end (17 January) of the 2016/2017 winter. We collected both leaf and lateral bud samples. The expression of *OeFT2* in leaf samples of Tzabar Kama ‘Barnea’ trees on January 17th are presented in Figure 6A. In Figure 9A, leaf *OeFT2* levels of these same samples are compared to samples from the earlier date, as well as to juvenile leaf samples from both dates. Clearly, as previously shown [19], *OeFT2* levels increase during winter in leaves of mature ‘Barnea’ trees (Figure 9A). Juvenile seedlings exposed to winter temperatures do not accumulate *OeFT2* transcripts in leaves (Figure 9A). AP2-like transcription factors, targeted by miR172, were shown to act as flowering repressors during the juvenile stage in some model plants, and a gene encoding a protein with an AP2 domain was reported to have higher levels in olive juvenile seedlings [16]. We scanned the olive genome [14] for genes encoding proteins similar to Arabidopsis AP2 (Appendix A), identified 13 genes encoding proteins similar to Arabidopsis AP2 or TARGET OF EARLY ACTIVATION TAGGED (TOE1), aligned them (Appendix A), defined two subgroups with the strongest homology (AP2_1; AP2_2, Table 2) and designed consensus primers to study gene expression of each group (Appendix A; Table 2). The joint expression of the AP2_1 subgroup (OE6A031451T1, OE6A037406T2, OE6A055418T1 and OE6A105872T1; Table 2) did not show significant differences between juvenile and non-juvenile tissue (Figure 9B). On the other hand, the joint expression of the AP2_2 subgroup (OE6A061030, OE6A068128, OE6A099997 and OE6A079258; Table 2), identified higher expression in juvenile trees, compared to adult trees in mid-January (Figure 9C). Interestingly, in mid-November, before flower induction, the combined expression of these four genes was not higher in juvenile seedling leaves, compared to leaves from mature trees (Figure 9C). Using gene-specific primers, we confirmed that two (OE6A061030 and OE6A099997; Table 2) of the four genes in the AP2_2 group did indeed have higher expression in January leaf samples in juvenile seedlings, compared to mature plants, while one of the genes (OE6A079258) did not (Appendix A). Finally, we studied the expression of *OeTFL1-1* in lateral buds in these samples (Figure 9D). Previously, the expression of this gene in buds in mid-January was found to be higher in trees previously carrying a heavy fruit load [19]. Here we found that expression of this gene in lateral buds of juvenile seedlings was much higher than in mature plants at both timepoints, with a much higher difference in mid-November, compared to mid-January (Figure 9D).

## 4. Discussion

We examined the effect of irrigation, cultivar and grove age on productivity of olive trees with an intermediate level of flowering. To estimate productivity, we collected information on 25 fruitless branches per tree. The average number of fruits per branch varied between 0 and 17.8 and seemed to correlate well with the total yield of the tree. Fruit numbers per branch depends on the number of new buds per branch, the percentage of buds forming inflorescences, the percentage of inflorescences bearing mature fruit and the mean number of fruits per fruit-bearing inflorescence. When we noticed a change in *x*, we determined the parameters that contributed to this change. In many cases, the product of *i* and *f* were well correlated with *x*, suggesting that changes in these two parameters had the biggest effect on *x*. The level of *i* can also be termed the degree of flower induction. Olive flower induction is affected by previous fruit load and cold winter temperatures in the winter [19]. Before those findings, it was proposed that winter chilling releases pre-formed olive floral buds from dormancy [38]. We show that levels of *OeFT2* gene expression in leaves towards the end of winter can provide an estimate on the levels of flower induction within a site. The level of *f* depends on the ratio of hermaphroditic flowers within an inflorescence, the degree of wind pollination from a compatible pollen source and weather conditions that either promote or prevent fruit set and development [39].

When comparing rain-fed to irrigated trees of several cultivars, the yield of irrigated trees is much higher [28]. The rate of flower induction, *i*, in the ‘Souri’ cultivar was the only parameter that was not significantly affected by irrigation regime (rainfed, as opposed to irrigation). It is reasonable to expect that rainfed groves do not suffer from drought in the characteristically rainy Israeli winter, when flower induction occurs [19]. In this case the almost two-fold increase in *x* in irrigated trees was likely due to changes in other parameters. The number of buds formed before winter and developmental events after anthesis (*f*, *m*) occur in the fall, late spring and summer when temperature rises and there is no rainfall. During these periods the water status of the rainfed groves is considerably lower than those receiving supplemental irrigation [40]. For example, in the fall, olive vegetative growth is dependent and correlated to the amount of water in the soil [41]. For this reason, the fall growth of the plants under the rainfed regime is suppressed and *n* is lower than that found in irrigated groves. Under rainfed conditions, drought during the fruit set and fruitlet development periods (late spring to summer) significantly reduced the percentage of inflorescences that bore fruit until maturity, as well as the number of fruits on such inflorescences. Both successful fertilization and the ability of the plant to ensure intact fruitlet development depend on environmental conditions [39].

When comparing groves under irrigation, the ‘Barnea’ cultivar reached a significantly higher level of *x* in comparison to the ‘Souri’ cultivar. The reproductive potential of the ‘Souri’ cultivar was actually higher up to anthesis. Branches in ‘Souri’ formed more buds, and the rate of flower induction was higher compared to ‘Barnea’, leading to a higher *k* value in ‘Souri’. However, because less inflorescences formed fruit (*f*), and such inflorescences carried less fruit (*m*), the *x* value was higher in ‘Barnea’. This difference between the cultivars was alluded to in recent work, comparing many cultivars and the number of fruits per inflorescences each had [42]. It may be that less productivity of inflorescences in ‘Souri’, compared to ‘Barnea’ is an inherent genotype-related trait, while the relative reduction in node number and flower induction in ‘Barnea’, in comparison to ‘Souri’, is due to the overall higher fruit load the ‘Barnea’ trees experience.

It is well known that without maintenance, older olive trees may reach a stage in which their capacity to produce sufficient one-year-old healthy shoots declines [26]. This is why pruning of older trees is conducted [27,28,29]. One of the hypotheses tested here was that the yields of irrigated intensive well pruned groves decrease with the increasing age of the grove. The concept of age in vegetatively propagated cultivars requires clarification. The original seedling of the ‘Barnea’ variety had a juvenile period in which it did not flower. After the transition to maturity, the propagation of this variety was through rooting cutting, as opposed to planting seeds, as this would have resulted in a wide variety of diverging genotypes. This means that all of the ‘Barnea’ trees in the world originated from the same seedling. Therefore, the age of an olive tree is usually calculated from the time that the cutting was taken for rooting. One of the groves of the ‘Souri’ cultivar used in this study was apparently planted hundreds of years ago, estimated by the local farmers as early as 1000 CE. This is of course more of a traditional estimate, and, based on published literature, likely several hundreds of years younger. Another parameter of the tree age is the part of the canopy that has one-year old branches. Flowering in the olive occurs in the spring on buds formed during the previous spring. Potentially a very old tree may have been heavily pruned over the years, such that most of its canopy is made up of one-year old branches, raising the question: what determines tree age? The results of this study show that the hypothesis that there is a yield reduction with age in well-maintained olive is incorrect. At some of the groves, there was no significant difference between the age groups; at some groves, the younger plot had a higher yield, while at others, the older plots actually had higher yields. It seems that in cases of yield reductions reported by olive growers, it is not the age of the groves causing lower yields but other factors that may have developed over time. One possibility, for example, is a lack of light penetration into the canopy of older trees. This will likely reduce flower induction, as well as the percentage of inflorescences bearing mature fruit [43].

As expected, the expression levels of the *OeFT2* gene in the leaf samples taken at the end of winter were positively correlated with the *i* value of the plot. The age of the vegetatively propagated groves did not affect *OeFT2* levels in new leaves. In contrast, the expression levels of *OeFT2* in juvenile seedlings at the end of winter were very low. The lack of FT accumulation in young seedlings is likely a major factor in olive juvenility. A similar finding was reported in mango [44]. In Arabidopsis, proteins from the AP2 family negatively control the expression of FT [12]. Previously the AP2 encoding unigene_5004 (currently annotated as OE6A119978T1) showed higher expression in apical shoot tips of branches from juvenile, compared to adult olives [16]. In most olive cultivars, inflorescences are formed in lateral buds, while apical buds remain vegetative. Gene expression in shoot tips may not fully represent the difference in flowering potential between juvenile and adult plants. Others [45] have studied the expression of a different AP2 encoding gene, termed contig_32154 (currently annotated as OE6A031451) in shoots of juvenile olive seedlings, showing an increase in expression from 3 to 6 months. Since FT is expressed in leaves, a juvenile suppressor of its expression would likely be expressed in leaves. Here, we studied expression in leaves of a subgroup (AP2_2) of AP2 encoding olive genes. Indeed, the combined expression of this subgroup was much higher towards the end of winter in juvenile seedlings, compared to mature trees. Two genes within this subgroup contributing to this difference were identified (OE60A9997 and OE6A061030). Another subgroup, containing the previously mentioned OE6A031451 gene, did not show increased expression in juvenile seedlings. It is still possible that, within this group, one or more of the genes shows increased expression in juvenile tissue. The increased expression of AP2-like proteins in leaves of mid-January juvenile seedlings may cause lower levels of *OeFT2* in these tissues. High levels of relative expression were also found for the *OeTFL1-1* gene in the buds of juvenile olive trees. In Arabidopsis, the TFL1 protein accumulates in the bud and competes with the FT protein, which is translated in the leaf and then moves to the bud, for protein–protein interactions, such as the one that takes place with the protein FD [46]. Despite the great similarity between the FT and TFL1 proteins, they have opposite effects on the flowering pathway with TFL1 inhibiting the process of the meristem differentiating into an inflorescence [47]. With the expression level of *OeFT2* being so low in the leaves of juvenile trees it is quite likely that the amount of FT2 protein that reaches the buds is negligible. Together with high levels of TFL1 in buds, this would ensure the inhibition of flowering in juvenile trees. Transgenic olives that constitutively overexpress the *Medicago truncatula FT* gene flower all year round and in juvenile seedlings [19]. If these transgenic trees have the same levels of TFL1 as non-transgenic trees (not tested), it would suggest that raising FT levels over a certain level in buds may be enough to overcome TFL1 competition and allow flowering of juvenile olive seedlings.

## 5. Conclusions

Our data suggest that well-maintained olive orchards do not lose productivity with age. The main parameters affecting final fruit numbers on trees were *i* and *f*, suggesting that the degree of flower induction, as well as the degree of successful fruitlet formation and persistence are the main factors affecting yield. As expected, previous fruit load reduced *i* levels. Irrigated ‘Souri’ had *f* values that were significantly higher than rainfed ‘Souri’ and significantly lower compared to irrigated ‘Barnea’, suggesting that this parameter is affected by both irrigation and genotype. Juvenile olive seedlings have higher levels of some *AP2-like* genes in leaves, together with lower levels of *OeFT2* expression. This, together with high levels of *OeTFL1-1* in buds, may be the mechanism to eliminate flowering in juvenile olives.

## Figures and Tables

**Figure 1 plants-11-02414-f001:**
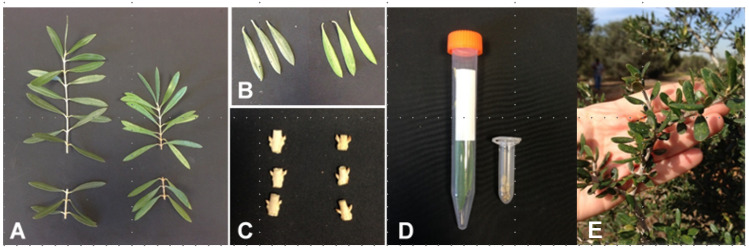
Tissue sampling for RNA extraction. (**A**) Branch samples were made up of the basal nodes of the new growth of two branches. (**B**) Three leaves were sampled from each branch. (**C**) Three nodes, made up of two axillary buds with a short section of stem between them, were sampled from each branch. (**D**) The six leaves and nodes were stored in closed tubes at −80 °C. (**E**) An example of a branch from a juvenile tree used for leaf sampling.

**Figure 2 plants-11-02414-f002:**
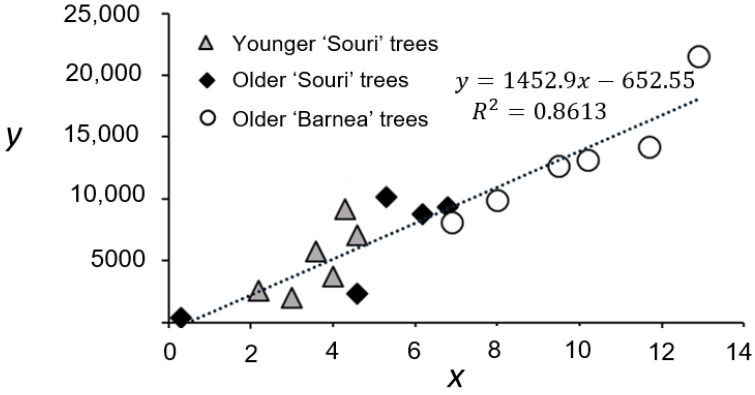
Correlation in final fruit number between a sampling of 25 random branches and a whole tree. Average fruit numbers per branch (*x*) based on 25 branches was compared to an estimate of fruit number for a whole tree (Y). The estimate relies on measuring weight of all fruit on the tree and measuring average fruit weight of fruits on each tree, based on a random sample of 30 fruits. Each dot represents a single tree, with altogether 17 trees representing three different groves (see legends) from Revivim 2018 harvest.

**Figure 3 plants-11-02414-f003:**
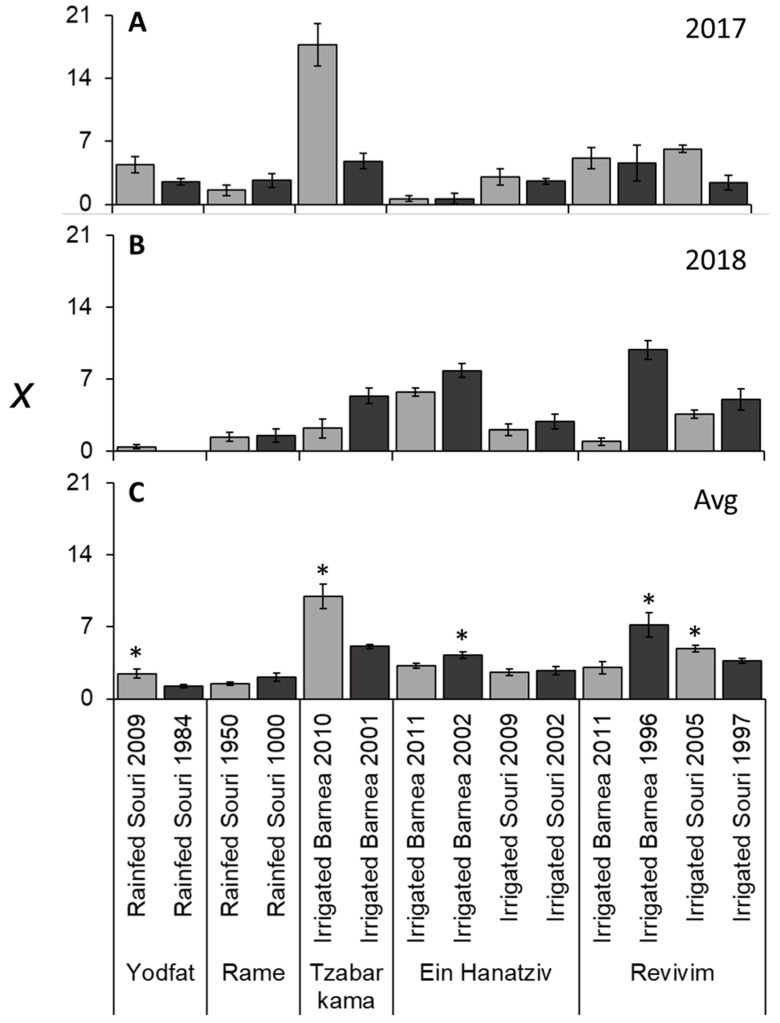
Number of fruits per branch (*x*) in different olive groves. First year (**A**), second year (**B**) and average (**C**) values presented. Values are averages of six trees per grove. See materials and methods for measurements. The error bars represent the standard error of the mean for each grove. In (**C**), asterisks mark significant differences between age groups in the same site (location, cultivar and irrigation method) calculated using the Student’s *t*-test (*p* ≤ 0.05).

**Figure 4 plants-11-02414-f004:**
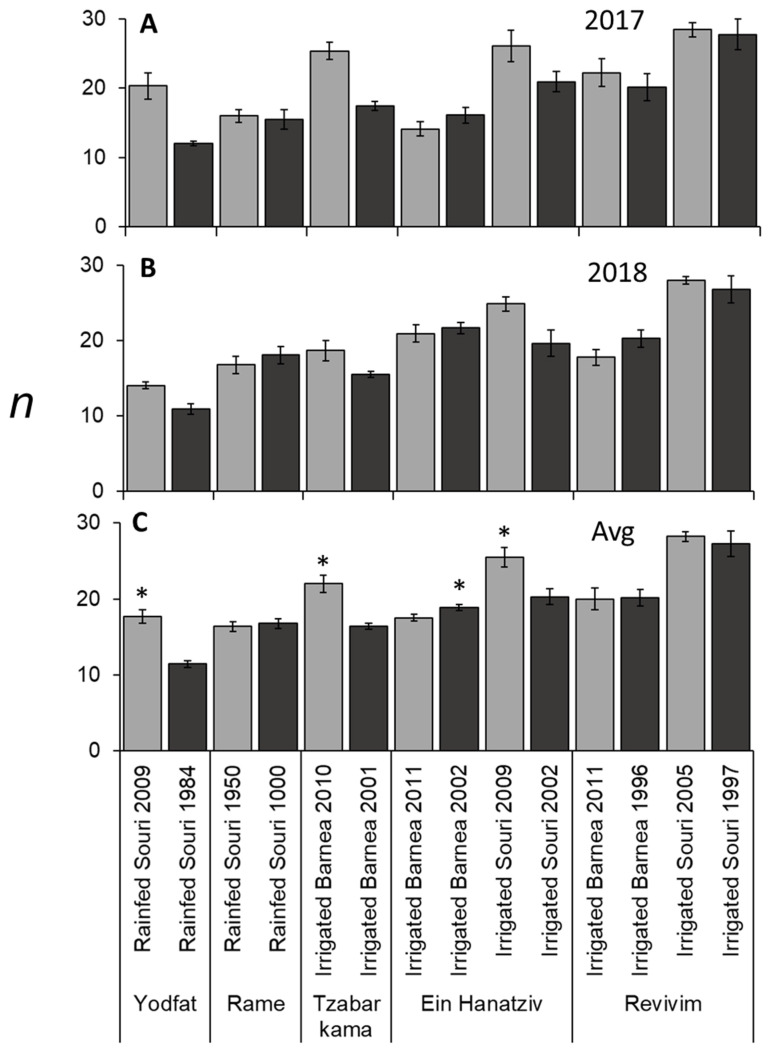
Number of nodes per branch (*n*) in different olive groves. First year (**A**), second year (**B**), and average (**C**) values presented. Values are averages of six trees per grove. See materials and methods for measurements. The error bars represent the standard error of the mean for each grove. In (**C**), asterisks mark significant differences between age groups in the same site (location, cultivar and irrigation method) calculated using the Student’s *t*-test (*p* ≤ 0.05).

**Figure 5 plants-11-02414-f005:**
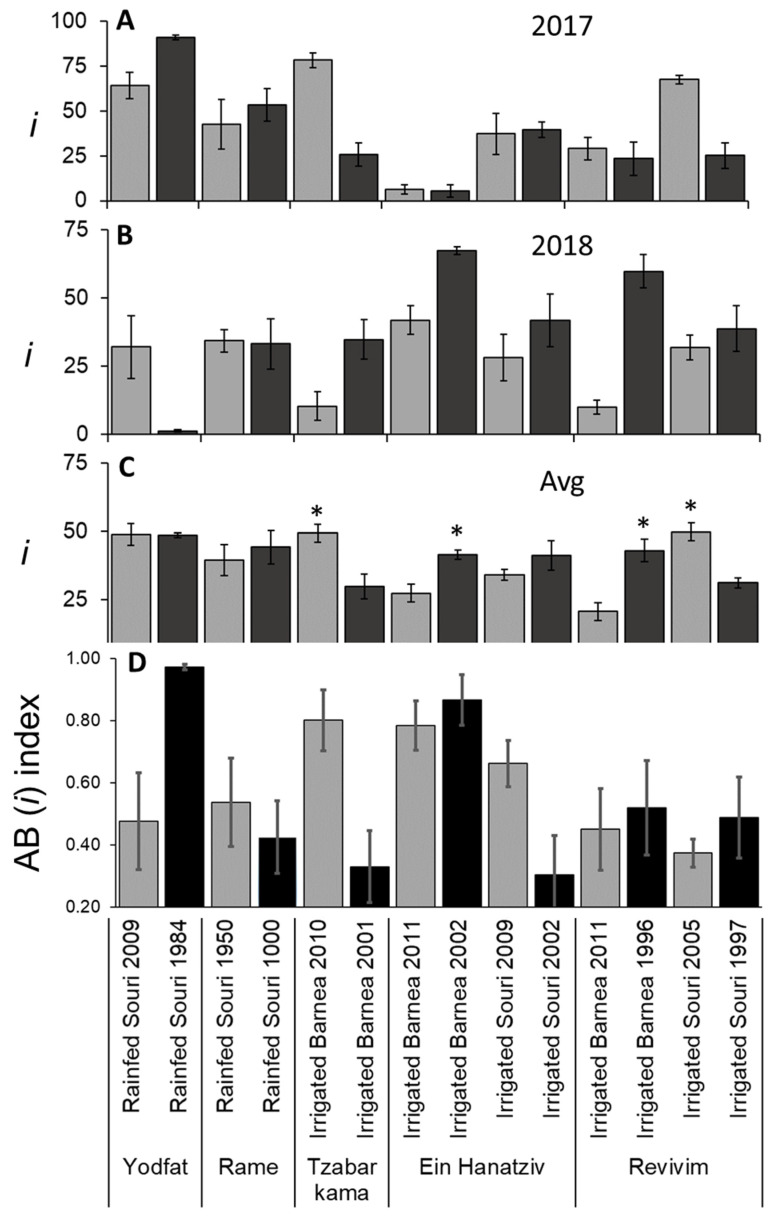
Percent nodes forming inflorescences (*i*) and alternate bearing (AB) index in different olive groves. First year (**A**), second year (**B**) and average (**C**) *i* values are presented. Values are averages of six trees per grove. (**D**) AB index calculation based on the first and second year *i* values. See materials and methods for calculation. The error bars represent the standard error of the mean for each grove. In (**C**), asterisks mark significant differences between age groups in the same site (location, cultivar and irrigation method) calculated using the Student’s *t*-test (*p* ≤ 0.05).

**Figure 6 plants-11-02414-f006:**
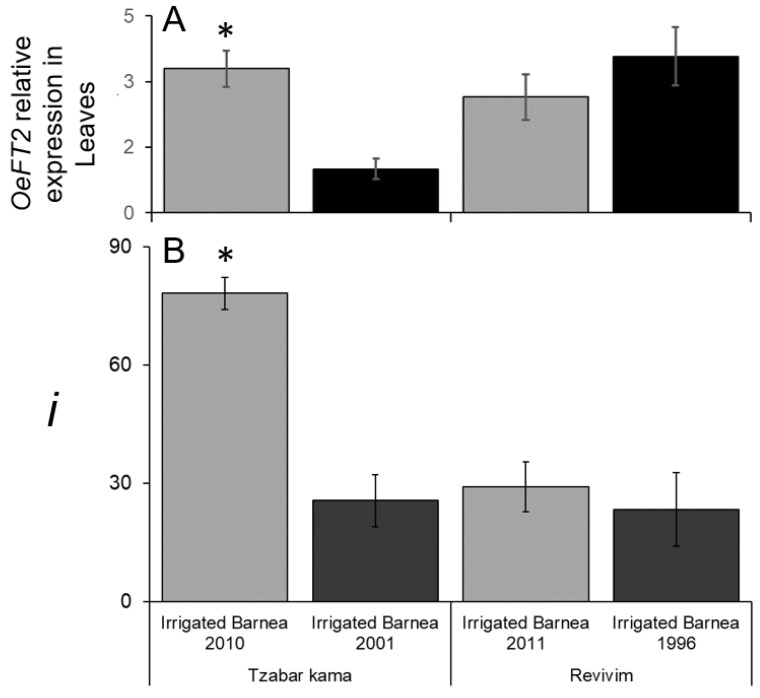
A comparison of *OeFT2* expression in leaves and percent nodes forming inflorescences (*i*) in four groves. (**A**) Comparison of *OeFT2* relative expression in leaves of ‘Barnea’ collected from four groves on the 17th (Tzabar Kama) or 19th (Revivim) January, 2017. Expression levels were measured using qPCR relative to the *OeACT7* gene. (**B**) Percent nodes forming inflorescences (*i*) in 2017 of orchards presented in (**A**). The same data for these orchards was presented in Figure 5A. Significant differences were calculated according to the Student’s *t*-test (*p* ≤ 0.05) with an asterisk showing a significant difference between young and mature trees at the same site.

**Figure 7 plants-11-02414-f007:**
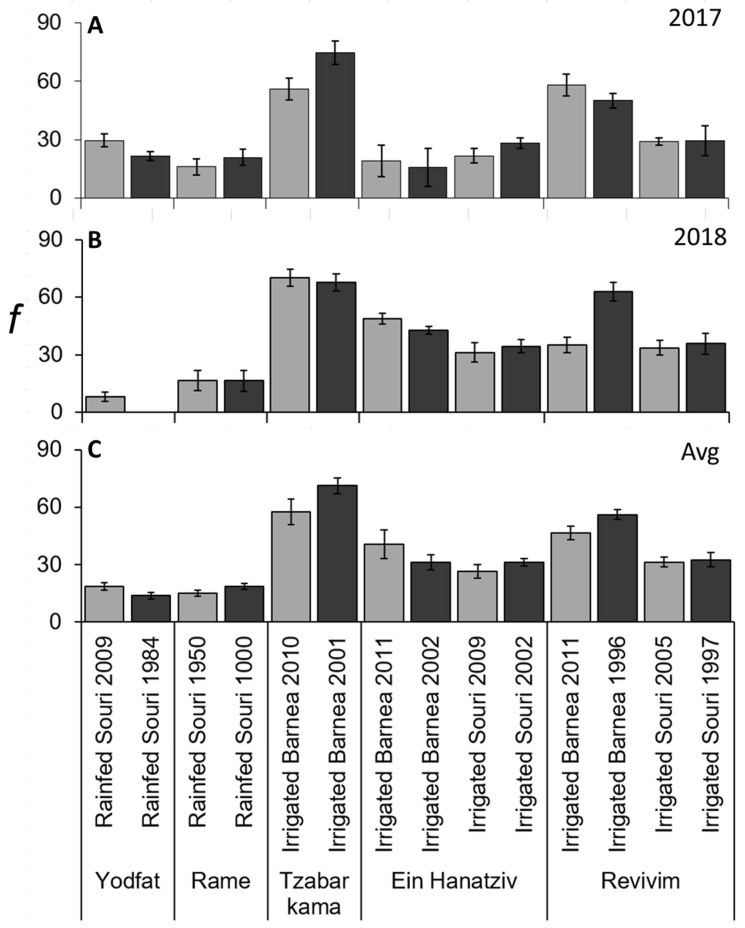
Percent inflorescences forming mature fruit (*f*) in different olive groves. First year (**A**), second year (**B**) and average (**C**) *f* values presented. Values are averages of six trees per grove. The error bars represent the standard error of the mean for each grove. No significant differences between age groups in the same site (location, cultivar and irrigation method) were identified using the Student’s *t*-test (*p* ≤ 0.05).

**Figure 8 plants-11-02414-f008:**
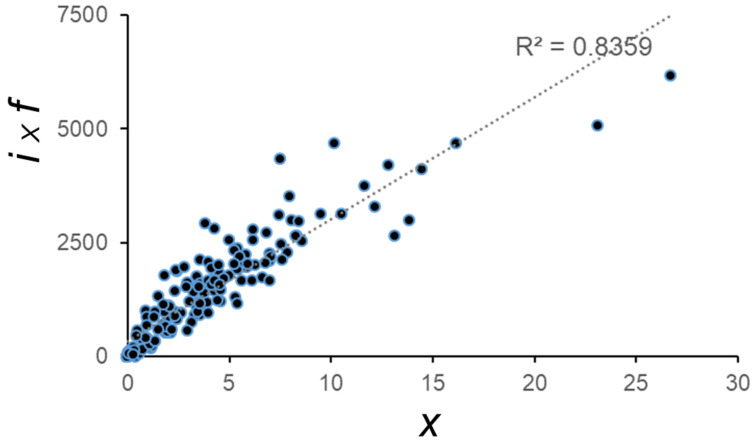
Correlation between the mean number of fruits per branch (*x*) and the product of multiplying the percentage of flowering buds (*i*) with the percentage of fruit forming inflorescences (*f*). The 168 data points are taken from both years of the experiment from 84 trees (each tree provided two data points, one per year).

**Figure 9 plants-11-02414-f009:**
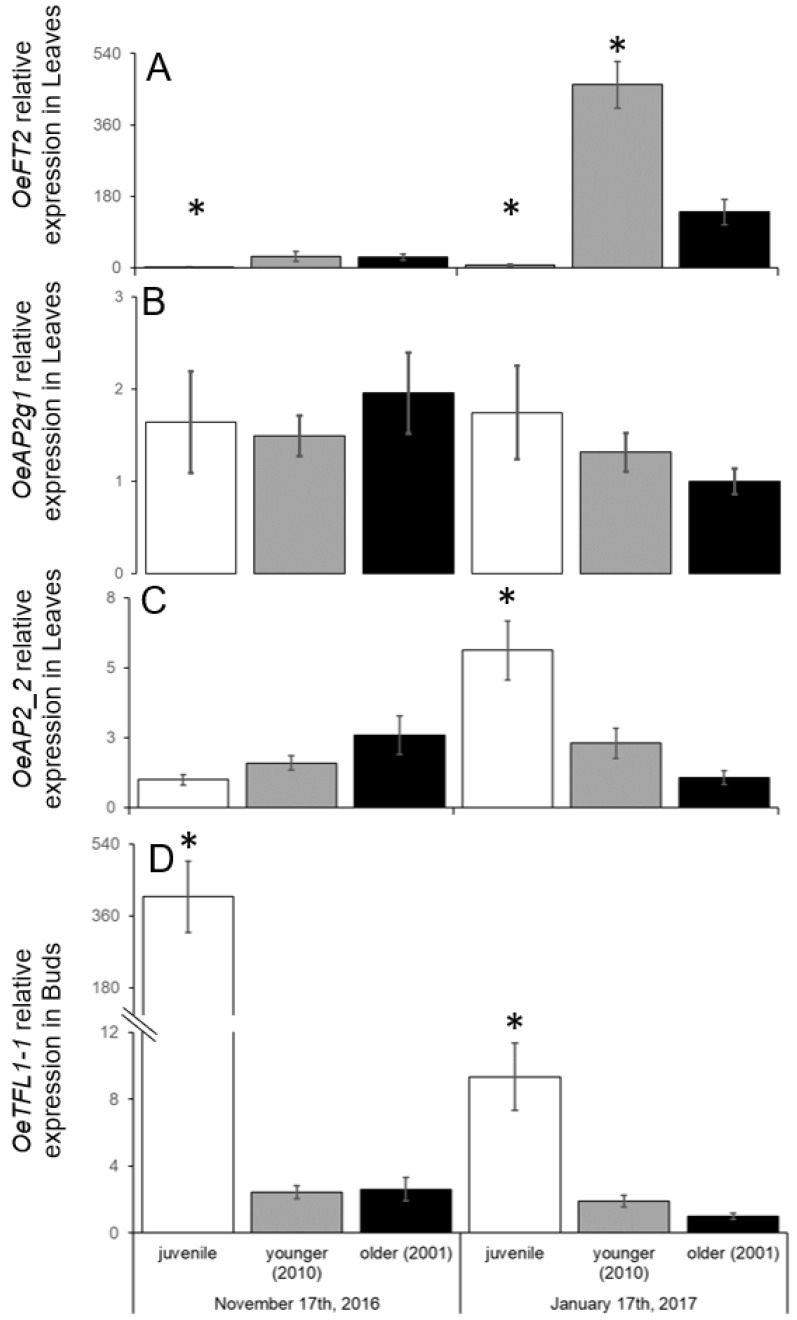
The effect of age and sampling date on expression of genes encoding flowering regulators. Leaf (**A**–**C**) and bud (**D**) samples were collected on 17 November 2016 and 17 January 2017 in a younger and older ‘Barnea’ orchard in Tzabar Kama, as well as in juvenile seedlings (‘Barnea’ as the female parent). Comparison of *OeFT2* (**A**), OeAP2_1 cluster (**B**) and OeAP2_2 cluster (**C**) relative expression in leaves and *OeTFL1-1* expression in buds (**D**). Data in (**A**) on younger and older orchards on January 17th is also included in Figure 6A. Significant differences in expression, compared to older trees within a specific date were calculated according to the Student’s *t*-test with an asterisk showing a significant difference (*p* ≤ 0.05).

**Table 1 plants-11-02414-t001:** Description of olive groves used in this study, including irrigation regime, region, cultivar and grove planting year.

Irrigation Regime	Location	Cultivar	Grove Planting Year	
			Older	Younger
Rain fed	Rame	Souri	1000 ^(a)^	1950
	Yodfat	Souri	1984	2009
Irrigated	Revivim	Barnea	1996	2011
		Souri	1997	2005
	Ein Hanatiziv ^(b)^	Barnea	2002	2011
		Souri	2002	2009
	Tzabar Kama	Barnea	2001	2010

^(a)^ Estimated by owner, likely several hundred years younger. ^(b)^ The younger groves in this location were in Ein Hanatziv yet belonged to Sde Eliyahu.

**Table 2 plants-11-02414-t002:** Primers for real-time RTPCR used in this study.

*Olea europaea*			Arabidopsis Reference Protein (from Tair)			Primers	
Name	Accession (Transcripts)	Genomic Location	Protein Name	Arabidopsis Accession	E-Value (Blastp)	Primer Name	Primer Sequence
OeACT7	OE6A117728T1	Oe6_s00163	ACTIN 7 (ACT7)	AT5G09810.1	5.6 × 10^−199^	OeACT7RTfor	5′-AAGATCAAAGTTGTTGCACCACC-3′
						OeACT7RTrev	5′-CTTAGAAATCCACATCTGCTGGAAT-3′
OeFT2	OE6A103537T1	Oe6_s04126	FLOWERING LOCUS T (FT)	AT1G65480.1	3.8 × 10^−74^	OeFT2RTfor	5′-CCTTCGTACTTTCTACACGCTCATT-3′
						OeFT2RTrev	5′-TCAGTCACCAACCAGTGCAAA-3′
OeTFL1-1	Oe6A037908T1	Oe6_s02173	TERMINAL FLOWER 1 (TFL1)	AT5G03840.1	3.2 × 10^−64^	OeTFL1.1RTfor	5′-CGTGAGTTCTGTCCGTCTGC-3′
						OeTFL1.1RTrev	5′-TCAGGATCAATCATCACCAGTGTA-3′
						OeTFL1.1Probe	5′-FAM-CCAGACCAAGGGTTGAGATTCAAGGAGGT-BHQ-1-3′
OeAP2_1 Group	OE6A031451T1	Oe6_s10134	TARGET OF EARLY ACTIVATION TAGGED (EAT) 1	AT2G28550.3	1.00 × 10^−^^101^	OeAP2_1_RTfor	5′-AACTTGGGCATTGCTCC-3′
	OE6A037406T2	Oe6_s07880		AT2G28550.3	9.00 × 10^−^^98^		
	OE6A055418T1	Oe6_s02174		AT2G28550.3	5.00 × 10^−^^105^	OeAP2_1_RTrev	5′-TTGAATTAGCGAATCCTGATG-3′
	OE6A105872T1	Oe6_s02170		AT2G28550.3	2.00 × 10^−103^		
OeAP2_2 Group	OE6A061030T2	Oe6_s06094	APETALA2 (AP2)	AT4G36920.1	5.76 × 10^−^^128^	OeAP2_2_RTfor	5′-GAAGAATTTGTGCATGTACTTCG-3′
	OE6A068128T1	Oe6_s00162			5.42 × 10^−^^125^		
	OE6A099997T2	Oe6_s07897			4.88 × 10^−^^123^	OeAP2_2_RTrev	5′-TATCATAAGCCCTGGCAGC-3′
	OE6A079258T1	Oe6_s07718			1.35 × 10^−^^63^		
OeAP2_2.1	OE6A099997T2	Oe6_s07897	APETALA2 (AP2)	AT4G36920.1	4.88 × 10^−^^123^	OeAP2_2.1_RTfor	5′-TGCCGAAGCACCAAGTGA-3′
						OeAP2_2.1_RTrev	5′-TTGGCTTAGGAGCTGCGTG-3′
OeAP2_2.3	OE6A079258T1	Oe6_s07718	APETALA2 (AP2)	AT4G36920.1	1.35 × 10^−^^63^	OeAP2_2.3_RTfor	5′-AGCTTAATTTCCAGAATGGACTTG-3′
						OeAP2_2.3_RTrev	5′-CCTACCGGTGAAGAGAACATT-3′
OeAP2_2.4	OE6A061030T2	Oe6_s06094	APETALA2 (AP2)	AT4G36920.1	5.76 × 10^−^^128^	OeAP2_2.4_RTfor	5′-GAAGTCTGCAGCGGCTGAG-3′
						OeAP2_2.4_RTrev	5′-GCTGTATCAAATCCACCCAA-3′

**Table 3 plants-11-02414-t003:** Mean values of parameters collected in this study.

Parameter	Irrigated		Souri	
	Barnea ^(a)^	Souri ^(b)^	Rainfed ^(c)^	Irrigated ^(d)^
Number of buds per branch (*n*)	19.17	25.31 ***	15.58	25.31 ***
Number of inflorescences per branch (*k*)	6.91	9.95 ***	7.22	9.95 **
Percent buds forming inflorescences (*i*)	32.70	38.70 *	45.00	38.70
Percent inflorescences that kept fruit (*f*)	50.61 ***	30.44	16.54	30.44 ***
Number of fruits on inflorescences that kept fruit (*m*)	1.55 ***	1.18	1.11	1.18 *
Number of fruits per branch (*x*)	5.47 ***	3.47	1.82	3.47 ***

^(a)^ Two-year average of data collected from six groves in three sites, six trees in each grove, twenty-five branches per tree. ^(b)^ Two-year average of data collected from four groves in two sites, six trees in each grove, twenty-five branches per tree. Notice the same data appears twice, each time in comparison to other cultivar or other irrigation regime. ^(c)^ Two-year average of data collected from four groves in two sites, six trees in each grove, twenty-five branches per tree. ^(d)^ Asterisks denote significant differences between irrigation treatments (* *p* ≤ 0.05, ** *p* ≤ 0.01, *** *p* ≤ 0.001) by Student’s *t*-test.

## Data Availability

See Appendix A.

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
