# Peer review of "Determining Reproductive Parameters, which Contribute to Variation in Yield of Olive Trees from Different Cultivars, Irrigation Regimes, Age and Location"

_plants, 2022, doi:10.3390/plants11182414_

Round 1

Reviewer 1 Report

Summary

This study tried to test whether there is a reduction in yield with increasing olive tree ages. Abundant data was collected. However, there was not a clear experimental design. The results presented in this study is not convincing.

In fact, in 2002, there was a study showing that yield of olive trees was affected by tree age evolution with a wide variation as initially increased from young age to mature but decreased after orchard was over 65 years old (Ben Rouina, B., Omri, A. and Trigui, A. (2002). EFFECT OF A HARD PRUNING ON TREES VIGOR AND YIELDS OF OLD OLIVE ORCHARDS. Acta Hortic. 586, 321-323

DOI: 10.17660/ActaHortic.2002.586.62). This is in consistent with many other fruit trees.

English in this manuscript needs to be improved. But language is not the main problem of this manuscript. I recommend authors make extensive revision on this manuscript, make the experimental design clear, delete predictions without enough supporting materials. For example, only focus on the 1 or 2 groves most data were collected from, to get more convincing experimental design and results.

General comments

1.       What is the experimental design?

2.       In introduction, please refer olive plant for a few more times. Take Lines 100-110 for instance, no olive tree / olive plant / olives were mentioned. It is confused for audiences reading this part. Similar problems found in lines 77-82.

3.       Table 1: Based on contents in lines 66-70, olive tree lifespan is between 200 and 900 years. However, the tree you studied in Rame orchard is over 1000 years old? You may add reference in introduction to give a wider lifespan estimate.

4.       Instead of using ‘~’, I recommend authors use ‘up to’.

5.       ‘five grams of mesocarp (pulp) were dried in oven at 90°C for 48 h and the dry weight was recorded’ in line 220. I doubted the 5 gram pulp can be fully dried at 90°C for 48 h.

6.       For the fruit size and oil percentage, plants were planted in different years and locations, I am not sure if the irrigation and fertilization management in two groves can be the same. As a result, the results might not be comparable.

7.       The equation used to calculate AB index is very confusing. What is the meaning of Year 1 and Year 2? Do you mean the flower number? Please explain.

8.       For the 1st part of Results, only one grove was studied to test the correlation between fruit number/tree and fruit number/branch. However, the pruning method, irrigation/fertilization management, shade, and many other parameters can affect this correlation. I am not convinced this correlation can represent yield in other olive trees and in other groves.

9.       For the equations between lines 300-305, as I discussed in general point 7, the correlation between X and Y may not be used. Thus, the equation/estimation may not be valid. Besides, for Table 3 & 4, how many groves were included? To be more specific, you should also mention how many branches/trees were included in the means of the data in Tables 3&4.

10.   Since there was abundant data presented in results, a lot of contents were included in discussion, I recommend that authors include a conclusion part.

11.   In the discussion part, more references should be included to compare your results with other studies, give possible reasons for some results, and so on. The current discussion is more like interpretation of results. Besides, some prediction is not rigorous, which should not be included in a scientific paper. For example, lines 565-567. I searched online, there are some studies evaluated the effect of age on olive tree yield.

Specific comments

Line 77: seeds

Line 78: The stage is termed as the ‘juvenile phase’.

Line 82-84: Please cite the original reference. Besides, ’12-13 years’ is different from ‘less than 13 years’.

Line 97-99: Please correct the citation format.

Lines 107 and 109: It will be better to mention ‘vegetative growth rate’ instead of ‘growth rate’.

Line 177: ‘Rameh’ or ‘Rame’?

Line 180: Dot after ‘ha’ should be removed.

Line 181: What about the spacing in Yodfat and Rame?

Line 190: ‘For’ instead of ‘From’

Reviewer 2 Report

The manuscript focus on an interesting and not yet clarified subject, flower induction and production in olive as influenced by different factors as tree age, cultivar and irrigation regime.

The overall manuscript is well written, introduction is consistent with the study, materials and methods clear and results and discussion well exposed.

Some minor correction are needed before publication as follows:

Line 77 “seedling cannot flower in the first years (up to 10-15 years)

Line 98 typo: the period mark is to be deleted.

Line 253: you should explain what ABS is. And which parameter of the year 1 and 2 can be used to calculate AB. Later in figure 7 you show the results from AB index calculation using two different parameters.

Line 272: I would write total number of fruits per tree (Y) instead of total yield.

Line 286 half instead of halve

Figure 3 charge a better resolution file, the present one is not very clear. In the graph the variety is called “Sori”, while in the text Souri (same thing in all the following figures)

Figure 7 the colours of the bars don’t correspond to the legend. The black (k older orchard) is dark gray in the bar graph.

Figure 8A it might be easier for the reader to have the A graph coupled with the %of flowering buds. Now you put all genes expression in one graph, but A has different dates and different groves of sample collection and is compared to figure 6A in text, thus I think it could be better to create a new graph with OeFT2 expression and % of flowering buds of the same orchards.

Figure 8 font is smaller then in the other figures either in the graph and in the capture.

Reviewer 3 Report

Dear authors,

The manuscript “Determining developmental stages that contribute to variation in yield of olive trees from different cultivars, irrigation regimes, age and location” is a study with scientific interest but as poorly organized and in some parts excessively extensive as in the results section that deconcentrates the reader from the content of the manuscript. Authors must make an effort to improve their manuscript before it can be considered for publication. I left a set of notes directly in the manuscript.

Best regards

Reviewer 4 Report

Abstract L52-54: this is well known from everybody, what did you find new?

L37 the number 14 groves is misleading. I understand that you measured in 7 farms with different age trees.

L48 in the rest of the manuscript you describe it as AP2. Why here and in the discussion in one point different name?

L169 You mean one irrigated and one rainfed? Or both irrigated the same way? Confusing paragraph. Only the table clarifies what you wanted to say.

L253 what is ABS, what values you used? Only the flowering density from the branches you measured? And the # of inflorescences is different than the # of flowers or the # of hermaphroditic flowers.

L281, 297 you present X and you describe it later …

In Figs. 3 and 4 the cv Souri is presented as Sori.

L184-188 you chose shoots per tree and then measured # of buds and inflorescences and fruits etc. You present buds per shoot and inflorescences per shoot. What about the variability in size of the shoots? The rainfed trees should have much smaller shoot growth. The length of the shoot could vary due to all variables studied. The expected calculation would be to present the data per some standard shoot unit, i.e. per 10 cm of shoot or per 50 cm shoot as examples. Expressing per shoot (and not branch, as branch is something much larger than the two year shoots you measured) puts a serious variability in the data.

L425-427Do not forget the effect of boron and nitrogen nutrition on fruit set and flower quality.

L522 what is this l or I? What is it described as?

L540 it is known now that flower induction starts, not even microscopically, in June, 11 months before flowering. Thus, the fall vegetative growth would be very difficult to bear flowers. If there is something else known on this matter (flowers developing on fall shoot growth), make a reference in the text.

L552-553 I do not understand the sentence. Both are smaller but fruit number was higher?

L605 and L597 the genes differ at the last number. What ‘previous’ do you mean?

References: non- uniform presentation of titles (with initial capitals or not)

Round 2

Reviewer 1 Report

Still, the experimental design (for example, randomized complete block design, split plot design, paired comparison design, etc.) was not mentioned in the new added part 3.1 in revised manuscript. Besides, design of the experiment, cultivation of the plants, and preliminary results that affected the design of the experiment, should be included in Materials and methods, instead of Results.

Cultivation management (such as cultivation, irrigation, fertilization methods) always should be included in materials and methods part in manuscripts of plant science. Please supplement details of related contents.

Reviewer 3 Report

Congratulations, I think you did a good job.

Author Response

Thank You!